

# Exploration of prognostic genes associated with lymphangiogenesis in breast cancer based on transcriptomics and experimental verification

Chen Liu[1], Tuo Zhang[2], Fushen Luo[2], Xiaofeng Yang[3], Yadong Li[1], Tonghui Yi[4], Shuang Wu[2], Yanbing Wang[5], Yueping Zhu[1] and Kun Zhao[3]

[1] Department of Clinical Laboratory, The Third Affiliated Hospital of Qiqihar Medical University, Qiqihar, Heilongjiang, China
[2] Department of Radiotherapy, The Third Affiliated Hospital of Qiqihar Medical University, Qiqihar, Heilongjiang, China
[3] Department of Clinical Pathology Diagnosis, Qiqihar Medical University, Qiqihar, Heilongjiang, China
[4] School of Medical Technology, Qiqihar Medical University, Qiqihar, Heilongjiang, China
[5] Department of Endocrinology, The Third Affiliated Hospital of Qiqihar Medical University, Qiqihar, Heilongjiang, China

Corresponding author
Chen Liu, liuchen1678@163.com

## ABSTRACT

**Background**. Breast cancer (BC), a malignant neoplasm resulting from the uncontrolled proliferation of mammary epithelial cells, is predominantly driven by pathogenic breast cancer gene (BRCA) 1/2 mutations in hereditary cases. Previous studies have implicated lymphangiogenesis in the progression of BC. This research aimed to identify prognostic genes associated with lymphangiogenesis in BC and explore their underlying biological mechanisms.

**Methods**. Publicly available datasets were utilized to identify differentially expressed genes (DEGs). Lymphangiogenesis-related genes (LRGs) were sourced from public databases, and candidate genes were determined through the intersection of DEGs and LRGs. Univariate Cox regression analysis and machine learning algorithms were employed to select prognostic genes and develop a prognostic model. Further analyses, including a nomogram, Gene Set Enrichment Analysis (GSEA), immune cell infiltration analysis, and drug sensitivity predictions, were conducted based on the identified prognostic genes. Finally, reverse transcription quantitative polymerase chain reaction (PCR) (RT-qPCR) was performed to evaluate the expression levels of these genes.

**Results**. By intersecting 9,577 DEGs with 179 LRGs, 109 candidate genes were identified. Ultimately, four prognostic genes—ZIC2, CD24, CEBPD, and CCL19—were selected, and a prognostic model was established. The model demonstrated robust performance upon evaluation and validation, with the nomogram confirming its strong predictive ability. Notably, the prognostic genes were found to influence pathways such as the cell cycle and EGFR ligands, as well as immune cells like activated CD4 T cells. Additionally, drugs like AUY922 and AZ628 showed considerable potential in treating BC. RT-qPCR results for these four genes in clinical samples aligned with the bioinformatics findings.

**Conclusion**. This study identified and validated four prognostic genes—ZIC2, CD24, CEBPD, and CCL19—that are associated with BC and may provide novel targets for diagnostic and therapeutic strategies.

## INTRODUCTION

Breast cancer (BC), a malignant neoplasm originating in the breast tissue, primarily results from the abnormal proliferation of mammary gland cells, leading to the formation of a tumor or mass. While pathogenic mutations in BRCA1 and BRCA2 are well-established genetic risk factors for hereditary BC, increasing evidence underscores the substantial role of non-hereditary factors, such as advanced age, familial predisposition, hormonal exposure (both endogenous and exogenous), lifestyle choices, and environmental carcinogens (*Obeagu & Obeagu, 2024*; *Singh et al., 2018*). BC is one of the most common malignancies, and in some regions, it has surpassed lung cancer in incidence rates (*Sung et al., 2021*). In China, the incidence of breast cancer has been increasing annually, likely due to changes in lifestyle, reproductive patterns, and improved diagnostic capabilities (*Lei et al., 2021*). Nevertheless, the mortality rate has declined in recent years, owing to the widespread adoption of early screening programs and advancements in therapeutic strategies (*Torre et al., 2016*). Furthermore, emerging treatments, including immunotherapy and PARP inhibitors, are showing promise in the management of BC (*Dvir, Giordano & Leone, 2024*; *Morganti et al., 2024*). Current therapeutic options for BC encompass breast-conserving surgery, mastectomy, radiotherapy, systemic chemotherapy, endocrine treatments, and molecularly targeted therapies. While many patients with BC experience improved prognoses compared to those with other solid tumors treated with multiple modalities, a subset still faces poor outcomes due to tumor heterogeneity. Thus, the identification of novel prognostic biomarkers remains critical for stratifying high-risk patients, providing insights for the development of targeted therapies, and improving clinical outcomes (*Houlahan et al., 2025*; *Moar et al., 2023*).

Lymphangiogenesis, the formation of lymphatic vessels, plays a key role in inflammation, wound healing, tumor progression, and metastasis (*Liu et al., 2023*). It has been well-documented that lymphangiogenesis is closely linked to lymphatic metastasis, distant metastasis, and adverse clinical outcomes across various cancers, including BC (*Song et al., 2024*). Lymphatic vessels also constitute a critical component of the immune microenvironment, influencing immune cell trafficking and viability, as well as affecting the efficacy of immunotherapies (*Deng et al., 2023*; *Ju et al., 2024*). As a complex process regulated by multiple factors and genes, lymphangiogenesis-related genes (LRGs) may serve as potential prognostic biomarkers for human cancers.

Tumor lymphatic vessel density correlates positively with clinical stage and axillary lymph node metastasis (*Sethy et al., 2021*). As tumors progress, enhanced lymphangiogenesis facilitates the entry of cancer cells into the lymphatic circulation (*Rezzola et al., 2022*). Newly formed lymphatic vessels are primarily located in the peritumoral region, rather than within the tumor itself. This distribution pattern reflects the impact of the local microenvironment on lymphangiogenesis, highlighting the peritumoral area as a critical site for lymphatic vessel formation (*Fujimoto & Dieterich, 2021*). In addition to an increased number of

intratumoral lymphatic vessels, these vessels undergo significant functional changes. On one hand, their heightened permeability allows for greater fluid and cell passage; on the other hand, these abnormally structured vessels often lack normal valve functions, leading to disrupted lymphatic flow, which is associated with a poor prognosis (*Karakousi, Mudianto & Lund, 2024*; *Li et al., 2024*). Moreover, accumulating evidence highlights the essential role of lymphangiogenesis in BC metastasis (*Hou et al., 2021*), positioning its inhibition as a promising therapeutic strategy. In conclusion, lymphangiogenesis plays a pivotal role in BC progression.

This study aims to clarify the molecular mechanisms and prognostic implications of LRGs in BC. Transcriptomic data from public databases were utilized to identify candidate genes associated with lymphangiogenesis through differential gene expression analysis and univariate Cox regression. A prognostic model and nomogram were subsequently developed. Further analyses explored the relationships between these genes, clinicopathological features, the tumor immune microenvironment, and potential drug targeting. This research emphasizes the critical role of LRGs in BC prognosis, offering a molecular basis for the development of targeted therapies. Through integrated analysis of transcriptomic and clinical data, prognostic biomarkers were identified to improve risk stratification, and novel therapeutic targets were uncovered, facilitating the translation of molecular insights into clinical applications for personalized BC treatment.

## MATERIALS & METHODS

### Data collection

The RNA-sequencing dataset comprising 1,084 BC tissue samples and 112 normal control breast tissue samples with survival information was obtained from the University of California, Santa Cruz (UCSC)-Xena website (https://xenabrowser.net/datapages/), referred to as The Cancer Genome Atlas (TCGA)-BRCA dataset. Additionally, the GSE20685 dataset (platform: GPL570) was sourced from the Gene Expression Omnibus (GEO) (http://www.ncbi.nlm.nih.gov/geo/), encompassing breast tissue samples from 327 patients with BC with survival data. Furthermore, a total of 179 LRGs were retrieved from the Molecular Signatures Database (MSigDB) (http://www.gsea-msigdb.org/gsea/msigdb/) (*Chen et al., 2023a*) (Table S1). All data were downloaded on February 8, 2025.

### Functional interpretation of candidate genes and construction of the protein-protein interaction (PPI) network

Differentially expressed genes (DEGs) between the disease and normal control group samples in the TCGA-BRCA dataset were identified using the DESeq2 package (v 1.38.0) (*Love, Huber & Anders, 2014*), with selection criteria of |log2FoldChange (FC)| >0.5 and $p < 0.05$. Based on log2FC values, the top 10 genes with the greatest up- and down-regulation differences were selected, and a volcano plot was generated using the ggplot2 package (v 3.4.4) (*Gustavsson et al., 2022*). A heatmap of the top 10 up- and down-regulated genes was created using the ComplexHeatmap package (v 2.14.0) (*Gu, Eils & Schlesner, 2016*). To identify candidate genes associated with lymphangiogenesis in BC development, the VennDiagram package (v 1.7.3) (*Chen & Boutros, 2011*) was used to intersect the DEGs
with LRGs. Gene Ontology (GO) and Kyoto Encyclopedia of Genes and Genomes (KEGG) enrichment analyses were then performed on the candidate genes using the clusterProfiler package (v 4.7.1.003) (*Wu et al., 2021*) with a threshold of $p < 0.05$, to explore significantly enriched functional terms and pathways. To investigate the protein-protein interactions (PPIs) among the candidate genes, a PPI network was constructed using the Search Tool for the Retrieval of Interacting Genes (STRING) database (interaction score >0.9) (http://www.string-db.org/) and visualized using the Cytoscape package (v 3.8.2) (*Shannon et al., 2003*).

## Construction and validation of prognostic models

In the TCGA-BRCA dataset, which includes survival data for 1,084 BC samples, the proportional hazards (PH) assumption was initially tested for candidate genes using the Cox.zph function to assess collinearity ($p > 0.05$). The PH hypothesis test was used to verify whether candidate genes met the proportional hazards assumption (constant hazard ratio) of the proportional hazards model, ensuring the reliability of subsequent survival analysis results. Then, the survival package (v 3.5-3) (https://cran.r-project.org/package=survival) was employed to perform univariate Cox regression model analysis on the candidate genes that met the PH assumption test, to screen candidate genes associated with survival (hazard ratio (HR) $\neq 1$, $p < 0.01$). A forest plot was generated with the forestplot package (v 3.1.1) (*Nakagawa et al., 2021*) to visually display the analysis results. Subsequently, the Least Absolute Shrinkage and Selection Operator (Lasso) model was applied using the glmnet package (v 4.1-4) (*Friedman, Hastie & Tibshirani, 2010*) with 10-fold cross-validation (CV). Candidate genes whose regression coefficients did not shrink to 0 at the optimal Lambda were considered prognostic genes. A prognostic model was then constructed using these genes, with the following formula:

$$RiskScore = \sum_{i=1}^{n} \left( Coef_i \times X_i \right)$$

where Coef represents the regression coefficient, and $X$ denotes the expression level of each prognostic gene. Based on the optimal cut-off value, the samples were divided into high-risk (HRG) and low-risk (LRG) groups. The Kaplan–Meier (K-M) survival curve, generated using the survminer package (v 0.4.9) (https://CRAN.R-project.org/package=survminer), was used to compare survival rates between the two groups ($p < 0.05$). The prognostic model's accuracy in predicting survival at 1, 3, and 5 years was assessed using the receiver operating characteristic (ROC) curve, drawn with the survivalROC package (v 1.0.3.1) (*Heagerty, Lumley & Pepe, 2000*). The distribution of risk levels and sample survival rates across the two groups was visualized using the ggplot2 package (v 3.4.4), and an expression heatmap of the prognostic genes was created using the pheatmap package (v 1.0.12) (*Shi et al., 2023*). Finally, the robustness of the prognostic model was validated within the GSE20685 dataset.

## Screening of independent prognostic factors and nomogram construction

A nomogram was constructed by identifying and selecting independent prognostic factors. Initially, for 902 patients with complete survival data from the TCGA-BRCA dataset, univariate Cox regression analysis was performed on risk score, age, gender, clinical stage, T stage, M stage, and N stage using the survival package (v 3.5-3) (HR $\neq$ 1, $p < 0.05$). The PH assumption test ($p > 0.05$) was then conducted on factors with $p < 0.05$. Multivariate Cox regression analysis ($p < 0.05$) and PH assumption testing ($p > 0.05$) were subsequently carried out on the factors that met the PH assumption criteria. The factors that passed these analyses were considered independent prognostic factors. A nomogram was developed based on these independent prognostic factors using the rms package (v 6.5-0) (*Xu et al., 2023*). The accuracy of the nomogram in predicting survival rates for the 902 patients with BC was evaluated using a calibration curve. Additionally, the prognostic performance of the nomogram model was assessed by generating ROC curves (area under the curve (AUC) >0.7) for 1-, 3-, and 5-year survival using the survivalROC package (v 1.0.3.1).

## Analysis of differences in clinical characteristics

To investigate the variations in risk scores across different clinical subgroups, risk scores between HRG and LRG were compared within subgroups of age ($\leq 60$, >60), gender (male, female), clinical stage (1, 2, 3/4), T stage (1, 2, 3/4), N stage (0, 1, 2, 3), and M stage (0, 1) using the Wilcoxon test ($p < 0.05$). Additionally, the expression levels of prognostic genes across these clinical subgroups were compared ($p < 0.05$).

## Gene set enrichment analysis (GSEA)

To explore the relevant signaling pathways and potential biological mechanisms distinguishing the two risk groups, gene expression differences between HRG and LRG samples were analyzed using the DESeq2 package (v 1.38.0), with genes sorted by log2FC. Gene sets "h.all.v2023.2.Hs.symbols.gmt" and "c2.cp.kegg.v2023.1.Hs.symbols.gmt" were retrieved from the Molecular Signatures Database (MSigDB) (https://www.gsea-msigdb.org/gsea/msigdb/) as reference gene sets, and GSEA pathway analysis was performed using the clusterProfiler package (v 4.7.1.003) (|normalized enrichment score (NES)| >1.0, $p < 0.05$) (*Wu et al., 2021*). Additionally, to explore the biological functions of prognostic genes involved in disease development, the "c5.go.v7.4.symbols.gmt" gene set was used as the background for the GSEA analysis. Associations between each prognostic gene and other genes were computed in the TCGA-BRCA dataset, with genes sorted according to their correlation coefficients. GSEA was then performed on these genes ($p < 0.05$) using the clusterProfiler package (v 4.7.1.003) (*Wu et al., 2021*).

## Investigation of immune infiltration

To investigate whether there were differences in the immune microenvironment between HRG and LRG, immune infiltration analysis was performed. The occurrence of BC is closely linked to immune system activity. Initially, the enrichment scores for 28 immune infiltrating cell types (*Xu et al., 2023*) were determined in HRG and LRG with survival data using the single-sample gene set enrichment analysis (ssGSEA) algorithm within

the Genomic Variation Analysis (GSVA) package (v 1.46.0) (*Hanzelmann, Castelo & Guinney, 2013*). To examine differences in immune cell infiltration between HRG and LRG, the Wilcoxon test was applied ($p < 0.05$), and the results were visualized using the ggplot2 package (v 3.4.4). Additionally, correlations between differential immune cells and prognostic genes were analyzed using the cor function from the psych package (v 2.2.9) (*Batista et al., 2021*) (|correlation coefficient (cor)| >0.3, $p < 0.05$). The transcription levels of 49 immune checkpoint genes (*Xiang et al., 2024*) were compared between HRG and LRG using the rank sum test ($p < 0.05$), and box plots were generated using the ggplot2 package (v 3.4.4).

## Analysis of differences in immunotherapy pathways and immune cycles

The cancer immune cycle plays a pivotal role in cancer progression. To explore differences in immune cycle activity between HRG and LRG, cancer immune cycle pathways were extracted from the Tumor Immune Phenotype (TIP) database (http://biocc.hrbmu.edu.cn/TIP/). The ssGSEA was applied to HRG and LRG samples using the GSVA package (v 1.46.0), and the Wilcoxon test was used to assess differences in the activity of immune cycle pathways between the two groups ($p < 0.05$). Furthermore, to investigate variations in immunotherapy prediction pathways between HRG and LRG, ssGSEA was conducted on the samples from both groups, followed by a Wilcoxon test to compare the activity of immunotherapy prediction pathways ($p < 0.05$). Box plots were created using the ggplot2 package (v 3.4.4) to visualize the findings.

## Drug sensitivity analysis

Tumor drug sensitivity data were obtained from the Genomics of Drug Sensitivity in Cancer 2 (GDSC2) database (https://www.cancerrxgene.org/). In the TCGA-BRCA dataset, the pRRophetic package (v 0.5) (*Geeleher, Cox & Huang, 2014*) was used to calculate the half-maximal inhibitory concentration (IC50) values for 138 commonly prescribed chemotherapy drugs. The correlation between risk scores and drug IC50 values was analyzed using the psych package (v 2.2.9) (|cor| >0.3, $p < 0.05$), and a volcano plot was generated using the ggplot2 package (v 3.4.4). The top 10 drugs showing the strongest correlation with risk scores and IC50 values were selected, and the Wilcoxon test was employed to assess differences in IC50 values between the HRG and LRG ($p < 0.05$), with results visualized through box plots created in ggplot2.

## Expression analysis and the reverse transcription quantitative PCR (RT-qPCR)

To examine the expression of prognostic genes in BC samples *versus* control samples, the expression levels of these genes in BC tissue samples were compared to control samples using the Wilcoxon test on the TCGA-BRCA dataset ($p < 0.05$). The expression of prognostic genes was also validated in BC cell lines (MCF-7, ZR-75-1, and SK-BR-3, with three samples per cell line) and control cell lines (MCF-10A, with three samples) by RT-qPCR. Among them, MCF-7 is a cell line commonly used for estrogen receptor-positive BC (*Neve et al., 2006*). ZR-75-1 and SK-BR-3 represent different BC subtypes, respectively

(*Aekrungrueangkit et al., 2022*; *Chen et al., 2023b*). As a normal mammary epithelial cell line, MCF-10A serves as a typical control cell line (*Soule et al., 1990*). The human BC cell lines (MCF-7, ZR-75-1, SK-BR-3, and MCF-10A) were generously provided by Haixing Biotechnology Co., Ltd. (Suzhou, China). Expression differences among the samples were analyzed using the $t$-test ($p < 0.05$). Prognostic gene expression levels were quantified using the $2^{-\Delta\Delta Ct}$ method. Primers for the prognostic genes were synthesized by Sangon Biotech Co., Ltd. (Shanghai, China) (Table S2). Data and $p$-values were calculated using GraphPad Prism 10, with GAPDH as the internal reference gene for normalization.

## Statistical analysis

Statistical analyses were performed using R (v 4.2.2; *R Core Team, 2022*). Group comparisons were conducted using the Wilcoxon test, with $p$-values <0.05 considered statistically significant.

## RESULTS

### Enrichment and PPI network analysis in 109 candidate genes

Differential expression analysis in the TCGA-BRCA dataset revealed a total of 9,577 DEGs, including 5,977 up-regulated and 3,600 down-regulated genes in the BC group ($|log2FC|$ >0.5, $p < 0.05$) (Fig. 1A). A heatmap of the top 10 up- and down-regulated genes, sorted by log2FC, is presented (Fig. 1B). From the intersection of 179 LRGs and 9,577 DEGs, a total of 109 candidate genes were identified for further functional enrichment analysis (Fig. 1C). These 109 candidate genes were associated with 2,090 biological functions and pathways, including 1,955 biological processes (BPs), 37 cellular components (CCs), and 98 molecular functions (MFs) ($p < 0.05$). The top five enriched BPs included lymph vessel development, vessel morphogenesis, and epithelial cell proliferation. The top five enriched CCs were platelet alpha granule, external side of plasma membrane, and platelet alpha granule lumen. The top five enriched MFs involved receptor–ligand activity, signaling receptor activator activity, and integrin binding (Fig. 1D, Table S3). Additionally, 137 KEGG pathways were enriched, with the top 10 pathways including the PI3K-Akt signaling pathway, focal adhesion, proteoglycans in cancer, and Ras signaling pathway (Fig. 1E, Table S3). These results suggested that the candidate genes might be widely involved in key biological processes such as blood vessel development, cell proliferation, and signal transduction, and were closely associated with BC occurrence and development.

Furthermore, a PPI network comprising 69 nodes and 178 edges was constructed. Genes such as FGF2, KDR, ITGB1, GRB2, and PIK3CA showed frequent interactions with other genes. The proteins corresponding to 40 genes were isolated and not included in the network (Fig. 1F). This indicated key roles of FGF2, KDR et al. in the PPI network, while the isolated proteins warrant further investigation for potential functions.

### Construction of a prognostic model using CCL19, CEBPD, ZIC2 and CD24 as prognostic genes

Among 109 candidate genes, 98 fulfilled the requirements of the PH assumption ($p > 0.05$) (Table S4). Subsequently, univariate Cox analysis was performed on these

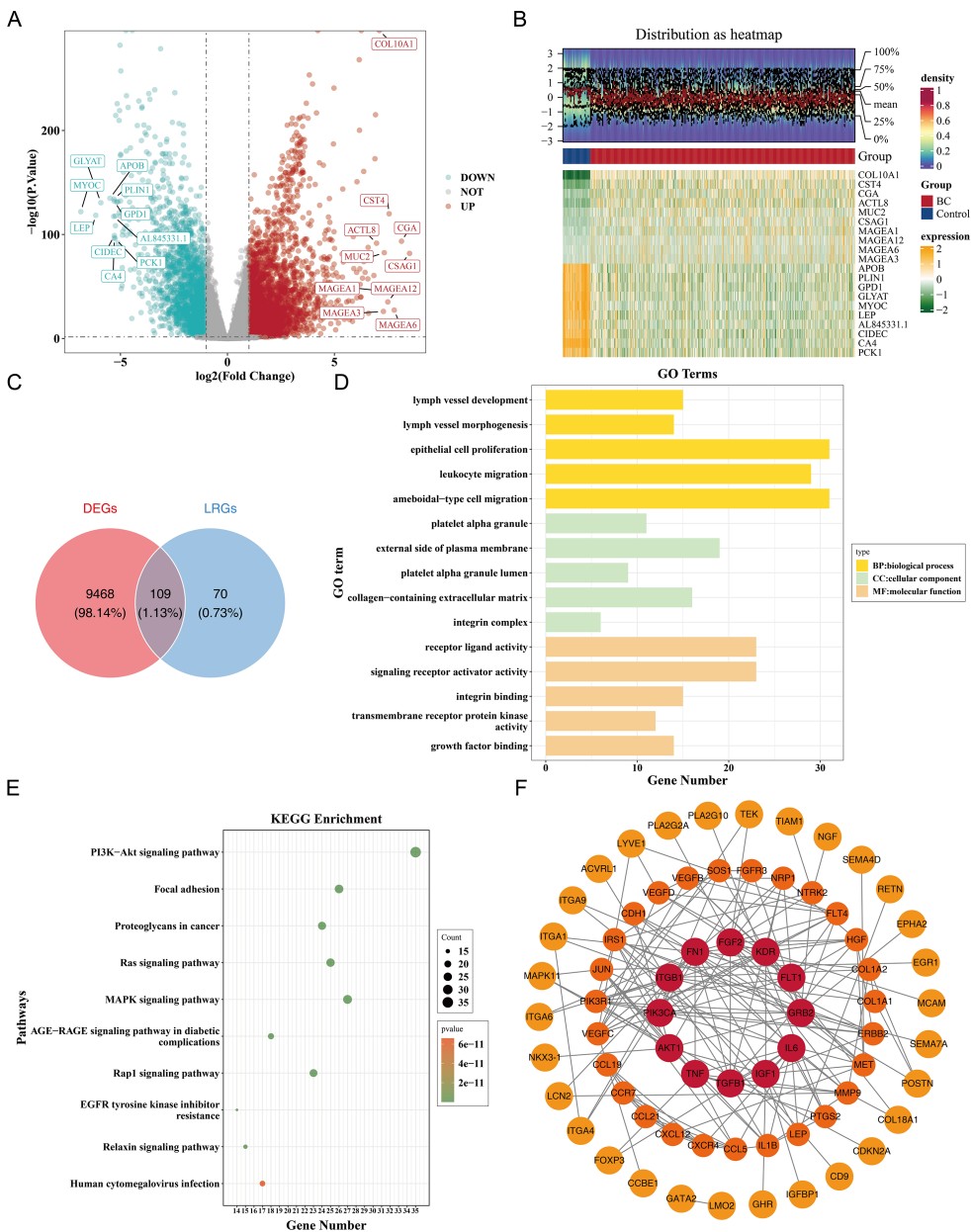

**Figure 1 Identification and functional enrichment of lymphangiogenesis-related candidate genes in breast cancer.** (A) Volcano plot of differentially expressed genes (DEGs) between breast cancer (BC) and normal tissues in the TCGA-BRCA dataset. Red and blue dots represent significantly up-regulated (5,977 genes) and down-regulated (3,600 genes) DEGs, respectively (|log2FC|> 0.5, $p < 0.05$). "DOWN" denoted down-regulation, "UP" denoted up-regulation, and "NOT" denoted no significant change in gene expression. (B) Heatmap of the top 10 up-regulated and down-regulated DEGs based on log2FC values. (C) Venn diagram illustrating the overlap between DEGs and lymphangiogenesis-related genes (LRGs), yielding 109 candidate genes. (D) GO enrichment analysis of candidate genes, highlighting key biological processes, cellular components, and molecular functions. (E) Kyoto Encyclopedia of Genes and Genomes (KEGG) pathway enrichment analysis, with the top 10 significantly enriched pathways (*e.g.*, PI3K-Akt signaling, focal adhesion). (F) Protein-protein interaction (PPI) network of candidate genes, with hub nodes indicating high connectivity. The redder the color is, the stronger the interaction between this gene and the rest of the genes at the protein level.

98 genes, and 4 prognosis-related genes (ZIC2, CD24, CEBPD, and CCL19) were identified ($p < 0.05$, HR $\neq$ 1) (Fig. 2A). In addition, based on the four prognosis-related genes, through Lasso regression analysis, it was found that when the optimal Lambda value was 0.001690118, the model was considered the optimal model, and four prognostic genes (ZIC2, CD24, CEBPD, and CCL19) were finally obtained (Figs. 2B–2C). The prognostic model was defined by the following formula: Risk score = (0.1758641) × CD24 + (−0.1753270) × CEBPD + (−0.0829096) × CCL19 + (0.1191970) × ZIC2. Using a cut-off value of 0.329995 for the risk score, TCGA-BRCA samples were classified into HRG ($n = 522$) and LRG ($n = 562$). The K-M curve showed that survival rates in the HRG were significantly lower compared to those in the LRG ($p < 0.0001$) (Fig. 2D). The ROC curve demonstrated the model's strong predictive capacity for the survival of patients with BC (AUC values: 0.646 at 1 year, 0.642 at 3 years, and 0.647 at 5 years) (Fig. 2E). Furthermore, as the risk score increased, the number of deaths also rose (Figs. 2F–2G). Heatmap analysis revealed high expression levels of CEBPD and CCL19 in the LRG, whereas CD24 and ZIC2 were more highly expressed in the HRG (Fig. 2H).

In the GSE20685 dataset, samples were similarly categorized into HRG ($n = 188$) and LRG ($n = 139$) based on the optimal risk score cut-off value of 0.1854183. Patients in the HRG exhibited significantly lower survival rates compared to the LRG ($p = 0.00046$) (Fig. 2D). The ROC curve again indicated that the predictive model was effective in estimating BC survival rates (AUC values: 0.632 at 1 year, 0.677 at 3 years, and 0.627 at 5 years) (Fig. 2E), with a corresponding increase in deaths as the risk score escalated (Figs. 2F–2G). Expression heatmap analysis showed that the expression patterns of the four prognostic genes in GSE20685 were consistent with those observed in TCGA-BRCA (Fig. 2H). These results validate the model's reliability and its potential utility in prognostic evaluation for patients with BC. The results of the GSE20685 dataset were consistent with those of the TCGA-BRCA dataset, indicating that the risk model constructed in this study was reliable.

## Nomogram constructed by four independent prognostic factors

Univariate Cox analysis (HR $\neq$ 1, $p < 0.05$) and the PH assumption test ($p > 0.05$) identified that the risk score, age, N, and M variables met the necessary criteria (Fig. 3A, Table S5). Subsequently, multivariate Cox regression analysis ($p < 0.05$) and PH assumption testing ($p > 0.05$) were conducted on these four factors. All four factors—risk score, age, N, and M—passed the corresponding analyses and were subsequently identified as independent prognostic factors (Fig. 3B, Table S6). To further verify the predictive effectiveness of independent prognostic factors for the 1, 3, and 5-years survival rates of BC patients, a predictive nomogram was developed (Fig. 3C). The calibration curves demonstrated that the slopes of the nomogram-predicted survival probabilities were close to 1 (Fig. 3D). This indicated that the nomogram model constructed in this study had good predictive accuracy. Additionally, the AUC values from the ROC curves (1-year = 0.72, 3-year = 0.73, and 5-year = 0.74) were all greater than 0.7 (Fig. 3E), further confirming the reliability of the constructed nomogram.

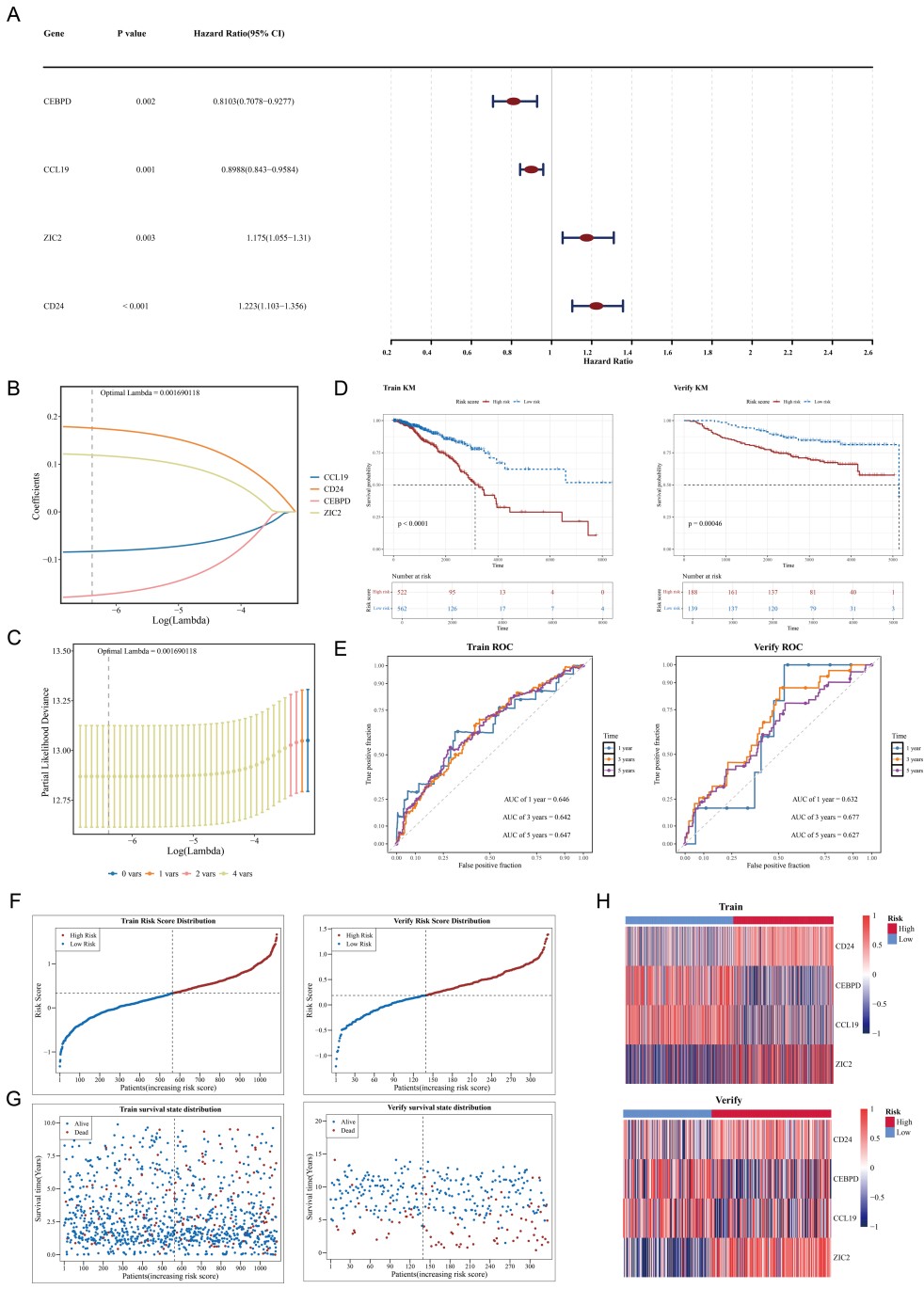

**Figure 2  Construction and validation of the prognostic model for breast cancer.** (A) Forest plot of univariate Cox regression analysis showing the hazard ratios of four prognostic genes (CEBPD, CCL19, ZIC2, CD24). (B) LASSO coefficient profiles of candidate genes. The optimal lambda (0.001690118) was selected *via* 10-fold cross-validation. (C) Partial likelihood deviance plot for LASSO regression. The vertical line indicates the optimal lambda with four nonzero coefficients. (D) Kaplan–Meier survival curves comparing high-risk and low-risk groups in the (Left) training set (TCGA-BRCA, *p* < 0.0001) and (Right) validation set (GSE20685, *p* = 0.00046). (E) Time-dependent ROC curves demonstrating the model's predictive accuracy at 1, 3, and 5 years (AUC: 0.646–0.647). (F) Risk score distribution and survival status of patients in the training set. (G) Risk score distribution and survival status of patients in the validation set. (H) Heatmap of prognostic gene expression patterns in high-risk *versus* low-risk groups.

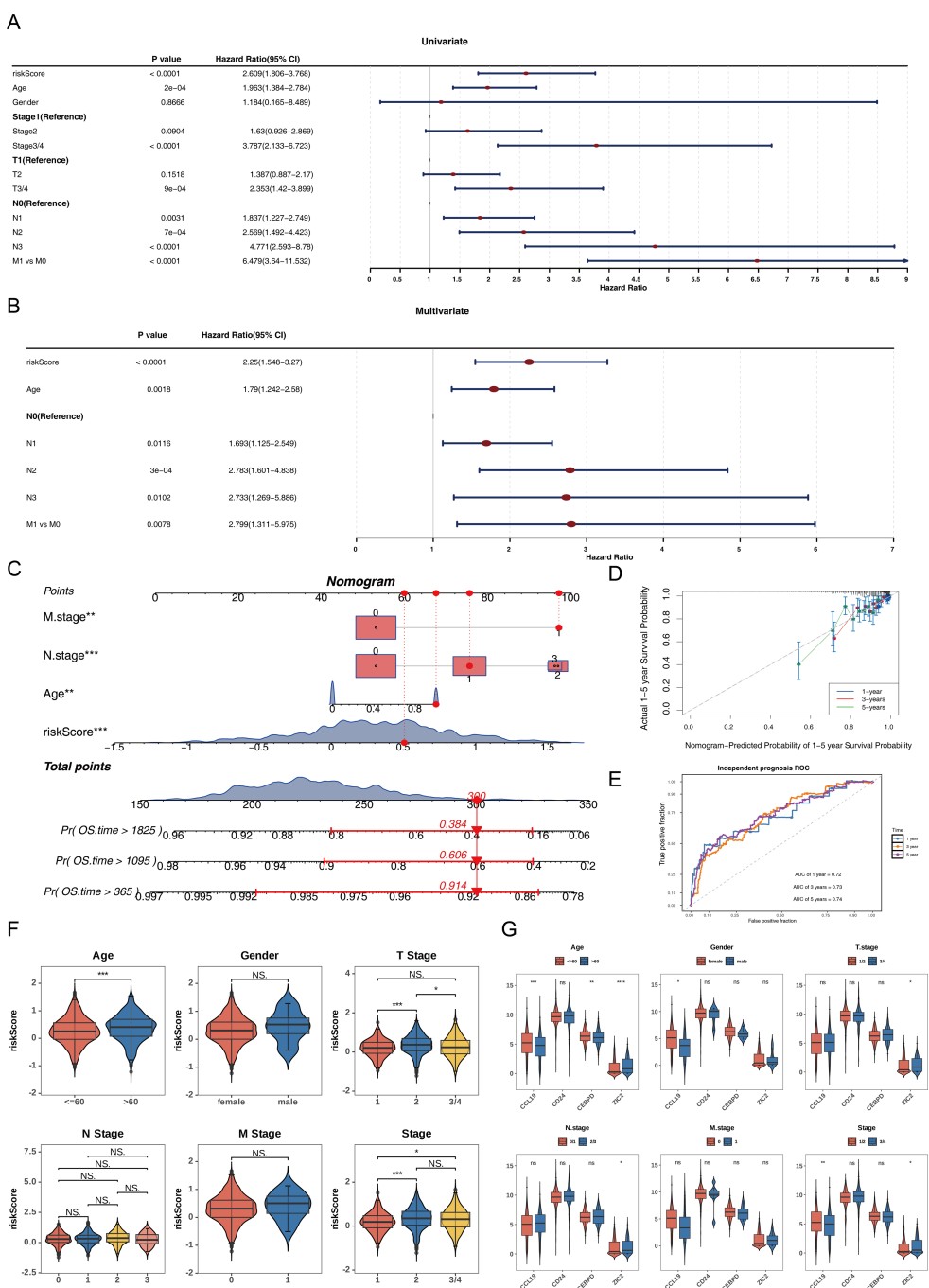

**Figure 3   Independent prognostic analysis and nomogram construction for breast cancer survival prediction.** (A) Univariate Cox regression analysis of clinical factors and risk score ($p < 0.0001$ for riskScore, age, stage, T/N/M status). (B) Multivariate Cox regression identifying independent prognostic factors: riskScore ($p < 0.0001$), age ($p = 0.0018$), and N/M stage ($p < 0.05$). (C) Nomogram integrating riskScore, age, and N/M stage to predict 1-, 3-, and 5-year overall survival probabilities. (D) Calibration curves showing agreement between nomogram-predicted and actual survival rates at 1/3/5 years. (E) ROC curves validating the nomogram's predictive accuracy (AUC: 0.72−0.74). (F) Risk score distribution across clinical subgroups (age $\leq$ 60 *vs.* >60, T1/2 *vs.* T3/4, stage1/2 *vs.* 3/4). (G) Differential expression of prognostic genes (CCL19, CEBPD, ZIC2, CD24) in clinical subgroups.

Among the subgroups defined by the six clinical characteristics, significant differences in risk scores were observed for age ($\leq 60$, $>60$), T (1/2), and clinical stage (1/2), with variations detected between the groups ($p < 0.05$) (Fig. 3F). Furthermore, CCL19 exhibited significant differences in expression across age, gender, and clinical stage (1/2 *vs.* 3/4), while CEBPD showed expression variations only in relation to age. ZIC2 demonstrated notable differences in expression levels across age, T, N, and clinical stage (1/2 *vs.* 3/4) ($p < 0.05$). CD24 did not display significant expression differences among any of the subgroups (Fig. 3G). These differences suggested that clinical characteristics such as age and tumor stage might affect the expression of prognostic genes, while CD24 might serve as a more stable potential marker.

## Functional enrichment pathways of HRG and LRG and prognostic genes

To elucidate the biological mechanisms of disease progression, the signaling pathways in BC were examined. Hallmark pathway analysis revealed 31 co-enriched pathways, and pathways such as G2/M checkpoint and E2F targets were significantly up-regulated in HRG (NES >1, $P < 0.05$), while allograft rejection, TNF-$\alpha$ signaling *via* NF-$\kappa$B, and interferon gamma response were significantly upregulated in LRG (NES<1, $P < 0.05$) (Fig. 4A). GSEA identified 50 co-enriched pathways, including hematopoietic cell lineage, cytokine-cytokine receptor interaction, and primary immunodeficiency, with significant up-regulation in the LRG ($p < 0.05$) (Fig. 4B). Additionally, pathway enrichment analysis for the four prognostic genes revealed that CCL19 was co-enriched in 2,641 pathways, CD24 in 1,530 pathways, CEBPD in 2,087 pathways, and ZIC2 in 985 pathways ($p < 0.05$). Notably, the four prognostic genes were significantly enriched in pathways related to the chromosome centromeric region, kinetochore, mitotic sister chromatid segregation, and chromosome segregation (Fig. 4C). These results indicated that significant differences existed between HRG and LRG in cell cycle, immune response, and chromosomal stability, suggesting that prognostic genes might influence BC progression by regulating these key pathways.

## Analysis of prognostic genes with distinct immune cells

The immune cell infiltration patterns are illustrated in Fig. 5A, where central memory CD4 T cells exhibited the highest level of immune infiltration between HRG and LRG. The Wilcoxon test showed that 24 types of immune cells were significantly down-regulated in HRG ($p < 0.05$), including activated CD4 T cells, activated dendritic cells, eosinophils, and natural killer (NK) T cells (Fig. 5B). The broad immunosuppressive state in HRG (such as down-regulated functions of activated CD4 T cells and natural killer T cells) might promote tumor immune escape, suggesting that immunotherapy might have limited efficacy in HRG patients. Moreover, most immune cells with differential infiltration demonstrated significant positive correlations with the prognostic genes. For instance, activated B cells showed a strong positive correlation with CCL19 ($p < 0.05$, $r = 0.8$) (Fig. 5C). Among 49 immune checkpoint genes, 39 exhibited significant differences between the HRG and LRG. For example, CCR7, BTN3A1, CCL5, CD200 and CD274 were all significantly highly

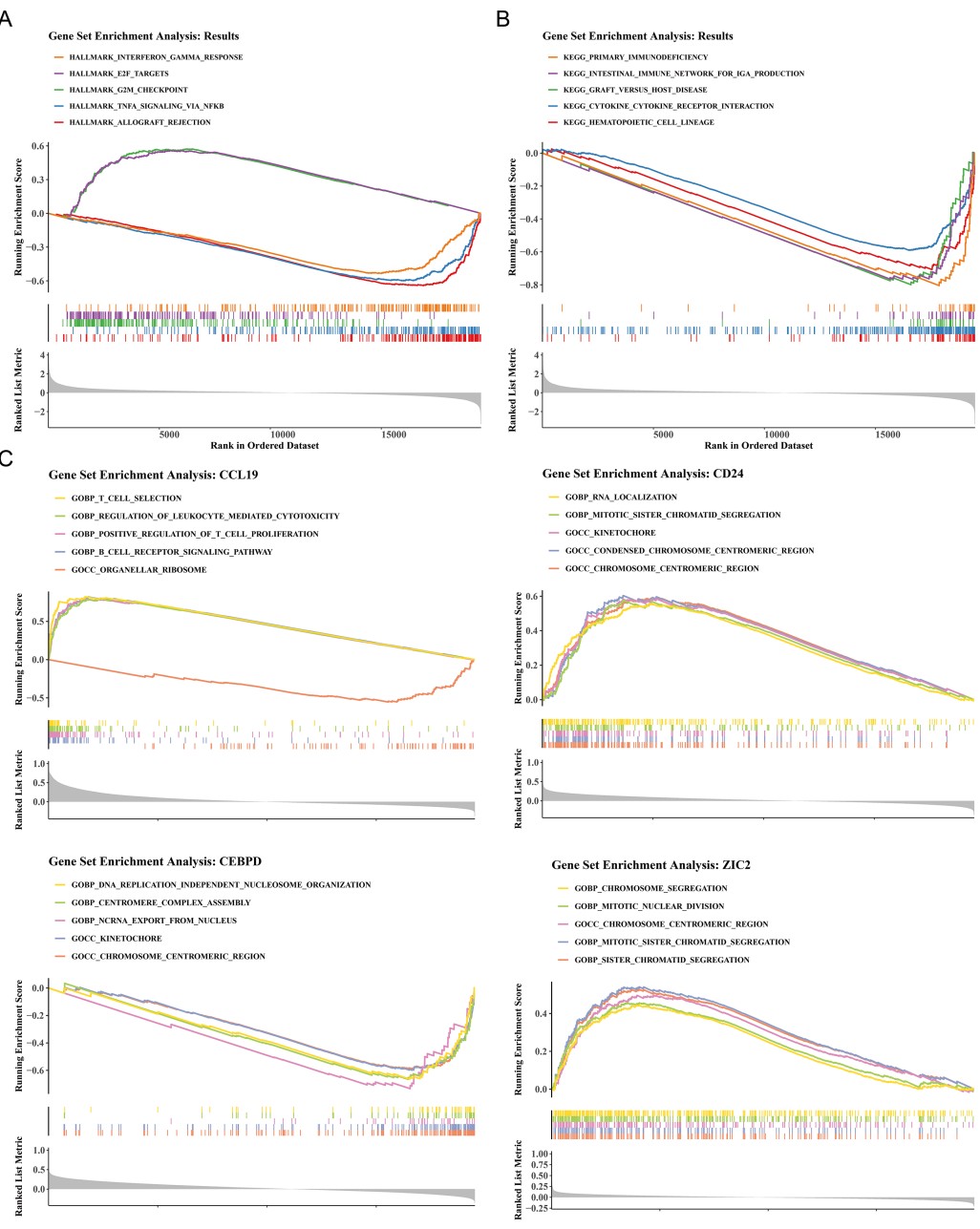

**Figure 4** **Functional enrichment analysis of high- *vs.* low-risk groups and prognostic genes.** (A) Hallmark pathway enrichment in risk groups. Top pathways: interferon-$\gamma$ response (NES = 1.82), E2F targets (NES = 1.79), G2/M checkpoint (NES = 1.77). (B) KEGG pathway enrichment. Key immune-related pathways: cytokine-cytokine receptor interaction (NES = 1.91), hematopoietic cell lineage (NES = 1.89). (C) GSEA of prognostic genes. The *X*-axis represents the "Rank in Ordered Dataset".

expressed in the low-risk group (Fig. 5D). Additionally, 22 of the 23 cancer immune cycle pathways showed significant differences between the HRG and LRG ($p < 0.01$), with the exception of the TH22 cell recruiting pathway in Step 4 (Fig. 5E). Similarly, among the 25 immunotherapy pathways, all but the hypoxia and catenin network pathways revealed

significant disparities between the HRG and LRG ($p < 0.05$) (Fig. 5G). These findings collectively reveal distinct immune landscapes and therapeutic implications between HRG and LRG.

## Correlation between risk score and IC50

The study also observed a correlation between the risk score and drug IC50 values. Among 138 drugs, the IC50 values of 26 drugs were associated with the risk score. Of these, PF.4708671 exhibited a negative correlation with the risk score (cor $= -0.325366603$, $p < 0.001$), while the remaining nine drugs showed a positive correlation (cor $> 0.3$, $p < 0.001$) (Fig. 6A, Table S7). The IC50 values of the top 10 drugs with the most significant correlations also demonstrated substantial differences between the HRG and LRG ($p < 0.001$) (Fig. 6B). These results suggest that the risk score could serve as a reference for personalized medication in HRG and LRG patients, with PF.4708671 being a promising therapeutic option for HRG patients.

## Expression analysis and experimental validation of CCL19, CEBPD, ZIC2 and CD24

In the TCGA-BRCA dataset, the expression levels of the prognostic genes CCL19, ZIC2, and CD24 were significantly upregulated in BC samples compared to normal samples, while the expression of CEBPD was notably downregulated in BC samples (Fig. 7A). RT-PCR results showed that the expression levels of CCL19, CD24, and ZIC2 were significantly higher in the MCF-7, ZR-75-1, and SK-BR-3 cell lines compared to the reference MCF-10A group, whereas the transcription level of CEBPD was significantly lower in the cancer cell lines ($p < 0.05$) (Fig. 7B). These gene transcription patterns align with the findings from bioinformatics analysis. The expression trends of prognostic genes obtained by RT-qPCR analysis were consistent with those of bioinformatics analysis. This indicated that the results of bioinformatics analysis were reliable.

## DISCUSSION

BC is a globally prevalent malignancy among women (*Culhane, Zaborowski & Hill, 2024*), with its heterogeneity contributing to poor prognosis in certain patients. Lymphangiogenesis plays a critical role in tumor metastasis and the immune microenvironment (*Zhang et al., 2025*). Using the TCGA-BRCA and GSE20685 datasets, this study identified four lymphangiogenesis-associated prognostic genes (CCL19, CEBPD, ZIC2, CD24), developed a risk model and nomogram, and validated their predictive capabilities. Functional enrichment and immune infiltration analyses highlighted distinctions between high- and low-risk groups, while RT-qPCR experiments confirmed the expression trends. These findings offer new perspectives for targeted BC therapy, though further investigation into clinical factors and drug mechanisms is warranted.

The CCL19 gene, located on chromosome 16q13 in humans, encodes C-C motif chemokine ligand 19, also known as macrophage inflammatory protein-3$\beta$ (MIP-3$\beta$). It is predominantly expressed in lymphoid and certain non-lymphoid tissues, where it plays a key role in recruiting immune cells and regulating immune responses (*Zhang et al.,*

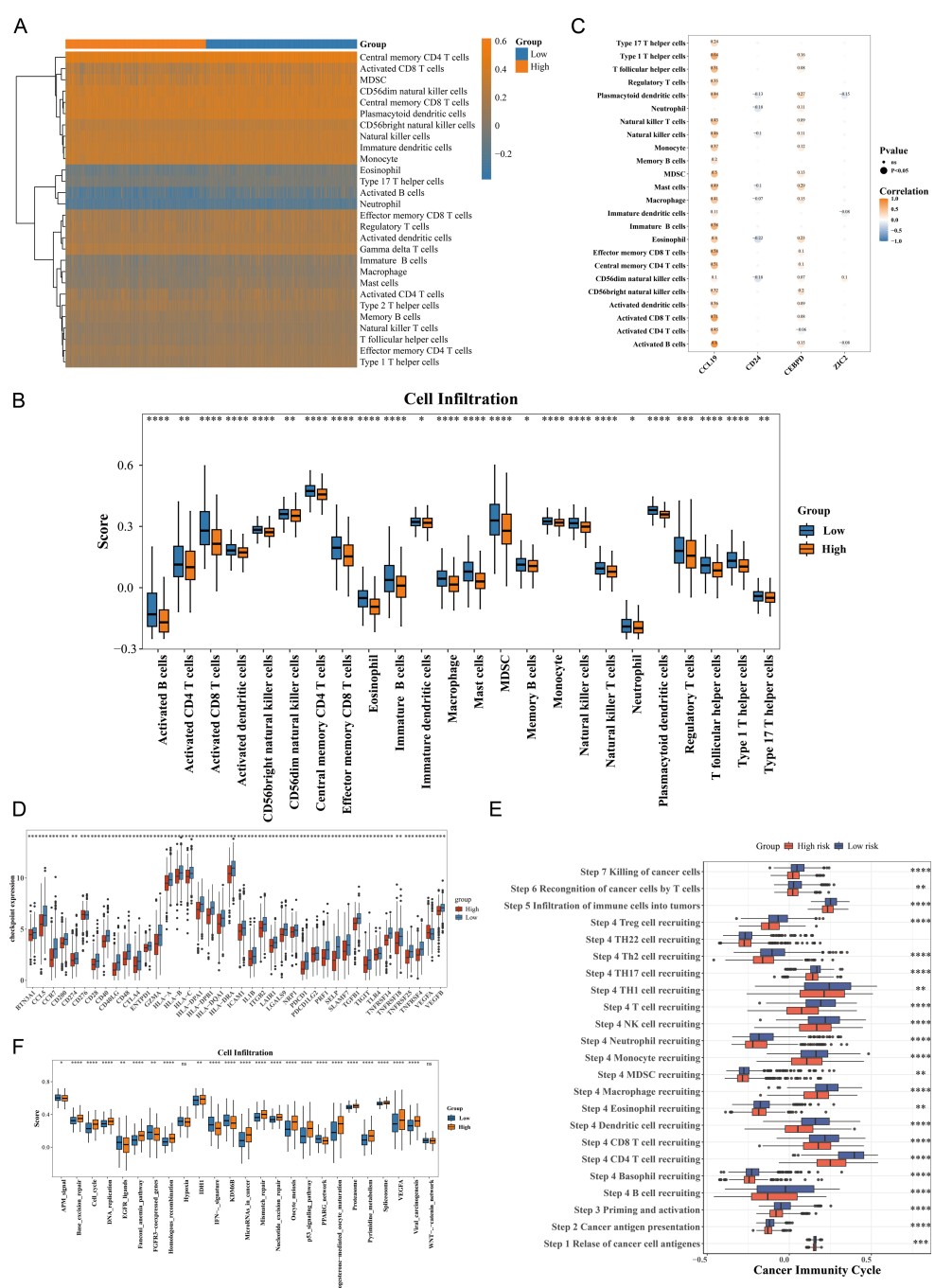

**Figure 5** **Immune microenvironment characterization and therapeutic implications in breast cancer risk groups.** (A) ssGSEA scores of 28 immune cell types in high- *vs.* low-risk groups. Central memory CD4+ T cells showed highest infiltration ($p < 0.001$). (B) Differential immune cell infiltration (24/28 cell types, $p < 0.05$), including activated CD4+ T cells and NK cells. (C) Correlation network between prognostic genes and immune cells. CCL19 strongly correlated with activated B cells ($r = 0.8$, $p < 0.001$). (D) Differential expression of 39/49 immune checkpoint genes ($p < 0.05$), including CD274 (PD-L1) and CTLA4. (E) Activity differences in 22/23 cancer immune cycle steps ($p < 0.01$), excluding TH22 recruitment. (F) Immunotherapy response pathways showing significant divergence (23/25 pathways, $p < 0.05$), particularly IFN-$\gamma$ signaling.

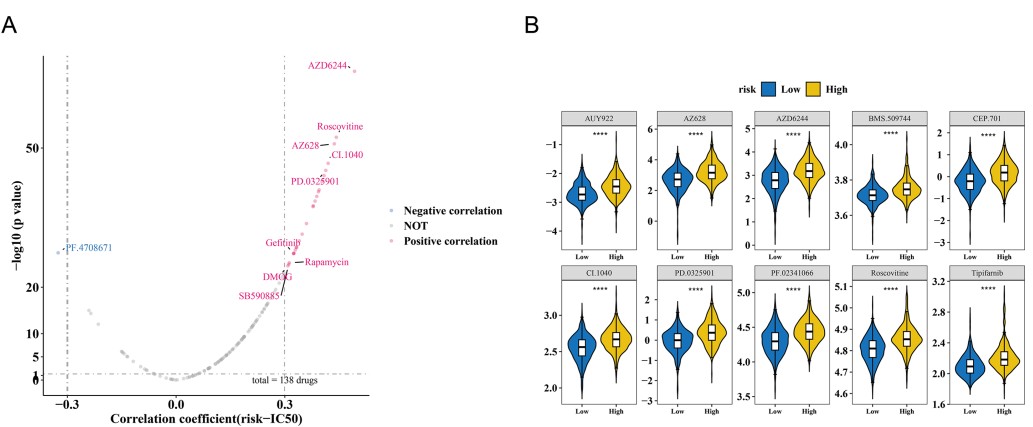

**Figure 6 Drug sensitivity analysis in breast cancer risk groups.** (A) Volcano plot of 138 drugs showing correlation between risk score and IC50 values ($|r| > 0.3$, $p < 0.05$). Top candidates: PF.4708671 ($r = -0.33$) and AZ628 ($r = 0.32$). (B) Boxplots comparing IC50 values of top 10 sensitive drugs between high- and low-risk groups ($p < 0.001$).

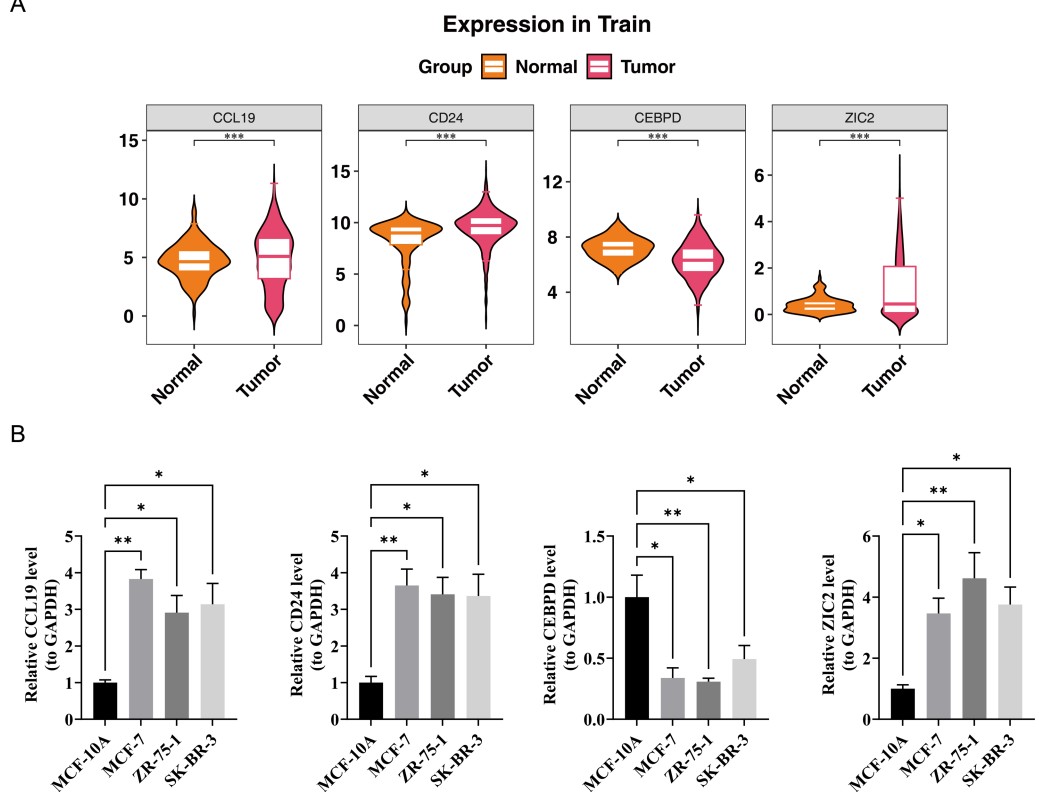

**Figure 7 Validation of prognostic gene expression patterns.** (A) Differential expression of CCL19, CD24, CEBPD and ZIC2 in BRCA *vs.* normal tissues from TCGA dataset (*$p < 0.05$, Wilcoxon test). (B) RT-qPCR validation in cell lines.

*2024*). In lymphangiogenesis, CCL19 binds to its receptor CCR7, activating downstream PI3K/AKT or ERK signaling pathways, thereby promoting the proliferation, migration, and tube formation of lymphatic endothelial cells (LECs) (*Wang et al., 2005*). In BC, CCL19, as a novel prognostic chemokine, exhibits aberrant high expression, which not only recruits immune cells to create a tumor microenvironment that inhibited for tumor growth and metastasis but also suppress tumor cell migration and invasion (*Gu et al., 2023*). Additionally, CCL19 augments lymphangiogenic potential by stimulating LEC sprouting and vessel maturation, facilitating tumor-associated lymphangiogenesis (*Korbecki et al., 2020*). Dysregulated expression of CCL19 and its receptor CCR7 has been widely implicated in various cancers. CCL19 recruits immune cells, including T cells and dendritic cells, to the tumor microenvironment *via* CCR7 binding, while promoting their accumulation in lymph nodes and other immune activation sites (*Sun et al., 2025*). High CCL19 expression correlates with improved prognosis and may predict response to PD-1/PD-L1 inhibitors, owing to the pre-activated immune microenvironment. Conversely, low CCL19 expression suggests immune evasion, necessitating combination therapies (*Murata et al., 2024*). Extensive research underscores the prognostic and immunomodulatory roles of CCL19 in BC. For example, *Hwang et al. (2016)* demonstrated CCL19's involvement in BC immune regulation by inducing dendritic cell chemotaxis. Moreover, CCL19 has been shown to modulate the BC immune microenvironment through interactions with tumor-infiltrating immune cells (TICs) (*Wang et al., 2022*). These findings align with the results of this study. Using the TCGA-BRCA dataset and RT-qPCR experiments, this study confirmed the elevated expression of CCL19 in BC samples and its association with immune cell infiltration through immune infiltration analysis. The results indicate that CCL19 is not only closely linked to immune cell infiltration in the tumor microenvironment, particularly in the recruitment of T cells and dendritic cells, but also strongly correlated with lymphangiogenesis. These findings highlight the dual role of CCL19 in regulating the tumor immune microenvironment and promoting lymphangiogenesis, offering new molecular insights into the mechanisms of lymphatic metastasis in BC. Thus, CCL19 could serve as a potential therapeutic target for intervention.

The CEBPD gene, located at 8p11.21, encodes a protein that belongs to the CCAAT/enhancer-binding protein (C/EBP) family of transcription factors. It is broadly expressed in various tissues and immune cells, where it plays a pivotal role in immune regulation and cell cycle control (*Hartl et al., 2024*). CEBPD inhibits the proliferation, migration, and invasion of cancer cells while promoting apoptosis (*Chan, Shiue & Li, 2023*). Recent research (*Zhao et al., 2025*) has identified CEBPD as an independent prognostic marker and a potential therapeutic target in BC based on bioinformatics analysis. However, in certain malignancies, aberrant overexpression of CEBPD has been linked to tumor progression and metastasis, potentially enhancing tumor cell survival and dissemination (*Mao et al., 2023*). Previous research (*Min et al., 2011*) has demonstrated that CEBPD promotes lymphangiogenesis and metastasis in lung cancer by regulating the HIF-1$\alpha$/VEGF-C signaling pathway. This finding not only supports our observation of CEBPD-mediated regulation of lymphangiogenesis in BC but also suggests that CEBPD may influence lymphangiogenic processes across various tumor types *via* conserved molecular pathways. In the analysis of the TCGA-BRCA dataset,

CEBPD expression was significantly downregulated in BC samples compared to normal tissues. This result was further corroborated by RT-qPCR experiments, which showed markedly reduced CEBPD transcription levels in MCF-7, ZR-75-1, and SK-BR-3 cell lines relative to the control MCF-10A cell line. The alignment between computational predictions and experimental data reinforces the potential of CEBPD as a reliable prognostic biomarker and a therapeutic target in BC, particularly in the context of lymphatic metastasis. These findings lay the groundwork for further exploration of CEBPD-mediated regulation of tumor-associated lymphangiogenesis and its clinical relevance in BC management.

The ZIC2 gene, located at 13q32.1, encodes a protein from the ZIC family of transcription factors. *Liu et al. (2020)* demonstrated that ZIC2 is downregulated in BC and suppresses tumor growth by modulating the STAT3 signaling pathway, positioning it as a potential prognostic biomarker for BC (*Liu et al., 2020*). Additionally, ZIC1 exerts tumor-suppressive effects by targeting survivin; increased expression of ZIC1 significantly inhibits tumor growth and downregulates survivin levels (*Han et al., 2018*). In BC, ZIC2 activates signaling pathways that accelerate cell cycle progression, promoting tumor cell proliferation. It may also regulate key molecules to help tumor cells bypass cell cycle checkpoints, leading to uncontrolled growth (*Lv et al., 2021*). *Yu et al. (2020)* identified the miR-129-5p/ZIC2 regulation of VEGF-C-mediated lymphangiogenesis in nasopharyngeal carcinoma. Our findings extend this mechanism to BC, where overexpression of ZIC2 correlates with lymphatic metastasis and poor prognosis, suggesting a conserved role in tumor lymphangiogenesis. Further research should focus on ZIC2's specific interactions with lymphatic endothelial cells within the breast tumor microenvironment. In the present study, analysis of the TCGA-BRCA training set revealed significant upregulation of ZIC2 in BC samples compared to normal tissues. RT-qPCR experiments confirmed markedly elevated ZIC2 expression in MCF-7, ZR-75-1, and SK-BR-3 cell lines relative to the control MCF-10A cell line. This alignment between experimental and bioinformatics data validates ZIC2 as a reliable biomarker for BC and suggests its involvement in tumor progression and lymphatic metastasis through modulation of the tumor microenvironment, particularly lymphangiogenesis.

The CD24 gene, located at 6q21, encodes the CD24 protein, a member of the sialomucin family. CD24 is widely expressed across various normal tissues, including hematopoietic, neural, and epithelial cells (*Panagiotou et al., 2022*). In BC and other malignancies, CD24 is often overexpressed, contributing to immune evasion, tumor cell proliferation, invasion, and metastasis (*Duex et al., 2017*). CD24 is also associated with cancer stem cell properties, which may drive tumor recurrence and resistance to therapy (*Tarhriz et al., 2019*). *Chan et al. (2019)* elucidated the role of the CD24-EGFR-STAT3 axis in driving lymphangiogenesis through VEGF-C upregulation. In the present study, analysis of the TCGA-BRCA training set revealed significant upregulation of CD24 in BC samples compared to normal tissues. This result was further validated by RT-qPCR, which showed markedly elevated CD24 expression in MCF-7, ZR-75-1, and SK-BR-3 cell lines relative to the control MCF-10A cell line. The consistency between experimental and bioinformatics analyses not only reinforces the reliability of these findings but also highlights the critical role of CD24 as a biomarker in BC. Furthermore, CD24 has been implicated in lymphangiogenesis, a key

process in tumor metastasis, suggesting its involvement in modulating lymphatic vessel formation and function within the tumor microenvironment. This dual role of CD24 in tumor progression and lymphangiogenesis underscores its significance as a potential therapeutic target for BC intervention.

A comprehensive analysis of significantly enriched pathways and their associated prognostic genes in high- and low-risk groups is essential for understanding disease mechanisms and guiding clinical treatment strategies (*Lei et al., 2024*). This study identified several significantly enriched pathways in tumor samples, including the G2/M checkpoint, E2F targets, cytokine-cytokine receptor interaction, chromosomal centromere region, and mitotic sister chromatid separation. These pathways are pivotal in tumorigenesis and progression, offering new insights into the molecular mechanisms of BC. Compared to previous studies, our findings further validate the significance of these pathways in BC. For instance, dysregulation of the G2/M checkpoint has been widely reported as closely associated with BC development (*Oshi et al., 2020*), and its significant enrichment in the high-risk group in this study further underscores its critical role in uncontrolled tumor cell proliferation. Similarly, the abnormal activation of the E2F targets pathway, identified as a driver of carcinogenesis in BC (*Suleman et al., 2019*), was also enriched in our study, highlighting its central role in tumor cell hyperproliferation and poor prognosis. Additionally, the cytokine-cytokine receptor interaction pathway, extensively studied for its role in the tumor microenvironment (*Wu et al., 2025*; *Xie et al., 2024*), was highlighted in our findings, emphasizing its importance in tumor cell proliferation, invasion, and immune evasion. The enrichment of pathways related to the chromosomal centromere region and mitotic sister chromatid separation aligns with existing literature, suggesting their potential roles in maintaining chromosomal stability and regulating BC progression (*Liu & Liu, 2022*; *Privette et al., 2008*). In summary, the findings of this study are highly consistent with previous research, further validating the critical roles of these pathways in BC.

This study revealed significant variations in 24 out of 28 immune cell types within the BC immune microenvironment. Correlation analyses demonstrated that most differentially infiltrated immune cells are strongly associated with prognostic genes, providing novel insights into BC pathogenesis and prognosis. For example, interactions between activated B cells, activated CD8 T cells, and CCL19 play a pivotal role in BC progression. Activated B cells, central to anti-tumor immunity, secrete antibodies targeting tumor cells. Elevated CCL19 levels, when associated with activated B cells, may facilitate their recruitment to the tumor microenvironment. These B cells can secrete cytokines such as interleukins, which influence tumor cell proliferation and immune evasion. A significant positive correlation between activated B cells and CCL19 was observed in this study, highlighting CCL19's potential role in modulating the tumor immune landscape. Activated CD8 T cells, vital effectors of anti-tumor immunity, directly kill tumor cells. As a chemokine, CCL19 directs activated CD8 T cells to migrate toward tumor sites. In BC, high CCL19 expression may augment the recruitment of activated CD8 T cells to the tumor microenvironment, enhancing immune surveillance and tumor cell eradication (*Comerford et al., 2013*). However, tumor cells may exploit various mechanisms to suppress the function of activated

CD8 T cells, contributing to immune escape (*Spranger et al., 2017*). The observed positive correlation between CCL19 and activated CD8 T cells in this study reflects the host's anti-tumor immune response, while also indicating that the balance between the tumor and immune system may be disrupted, potentially influencing BC progression. Significant differences in immune cell infiltration and immune checkpoint gene expression were observed between HRG and LRG. Elevated CCL19 expression in the LRG was associated with increased infiltration of activated CD8 T cells and dendritic cells, suggesting improved responsiveness to anti-PD-1/PD-L1 therapies. In contrast, HRG patients, characterized by limited immune cell infiltration, may benefit more from targeted therapies (*e.g.*, AUY922, AZ628) or combination immunotherapies. Future trials should utilize these immune signatures to guide personalized treatment strategies.

This study integrated OS data from BC cohorts with pathway enrichment analysis, revealing a significant discovery: eight immune-related pathways exhibiting distinct activity patterns between high-risk and low-risk groups. These pathways include recruitment of CD8 T cells, recruitment of NK cells, recruitment of Tregs, cancer antigen presentation, T cell recognition of cancer cells, Cell cycle regulation, EGFR ligands, IFN-$\gamma$ signature, and the PPARG network. The findings provide valuable insights into BC immunotherapy mechanisms and serve as a foundation for targeted therapeutic approaches. In BC, tumor cells employ various strategies for immune evasion, including overexpression of MHC class I molecules to impair NK cell function (*Zhang et al., 2022*), recruitment of Tregs to secrete IL-10 and TGF-$\beta$, thereby suppressing CD8 T cell and NK cell activity (*Jung et al., 2023*; *Luo et al., 2024*), and downregulation of antigen expression or interference with dendritic cell function to hinder T cell activation (*Yin et al., 2023*). Additionally, tumor cells exploit PD-L1 to suppress T cell recognition, while aberrant EGFR signaling and cell cycle dysregulation promote cell proliferation (*Glaviano et al., 2024*; *Saxena & Dwivedi, 2012*; *Vathiotis et al., 2022*). The dual role of IFN-$\gamma$ and the PPARG regulatory network suggest promising therapeutic targets (*Cao et al., 2025*; *Li et al., 2022*). Combined strategies targeting these mechanisms, such as immune checkpoint inhibitors, EGFR inhibitors, and PPARG activators, hold promise for enhancing immunotherapy efficacy and improving patient outcomes.

The prognostic genes associated with lymphangiogenesis identified in this study and the risk model constructed provide new insights into the prognostic assessment of breast cancer (BC). Lymphangiogenesis plays a crucial role in tumor progression and metastasis; therefore, these genes not only help elucidate the molecular mechanisms underlying BC metastasis but may also provide potential targets for clinical treatment (*Song et al., 2024*; *Zhang et al., 2023*). The risk model we developed can assist clinicians in assessing patients' prognostic risk at an early stage, providing guidance for personalized treatment. Through precise risk stratification, patients can select more appropriate treatment strategies based on their different prognostic risks, optimizing treatment outcomes and avoiding overtreatment. Additionally, further clinical validation and functional studies will help elucidate the specific roles of these prognostic genes in the lymphatic vessel formation process of breast cancer, driving the optimization of breast cancer treatment strategies and improvements in clinical practice.

## CONCLUSIONS

The findings of this study offer novel insights into the molecular mechanisms underlying BC, suggesting that targeting LRGs could represent a novel approach to enhancing prognostic assessment and therapeutic interventions. However, further validation in larger clinical cohorts, alongside mechanistic studies, is necessary to fully elucidate the functional roles of these genes and their potential as therapeutic targets. This research expands the understanding of BC heterogeneity and lays the groundwork for future precision medicine strategies.

## ACKNOWLEDGEMENTS

We would like to express our sincere gratitude to all individuals and organizations who supported and assisted us throughout this research. In conclusion, we extend our sincere gratitude to all individuals and institutions who supported this research.

### Funding
This study was supported by the Natural Science Foundation of Heilongjiang Province (No. PL2024H260). The funders had no role in study design, data collection and analysis, decision to publish, or preparation of the manuscript.

### Grant Disclosures
The following grant information was disclosed by the authors:
Natural Science Foundation of Heilongjiang Province: PL2024H260.

### Competing Interests
The authors declare there are no competing interests.

### Author Contributions
- Chen Liu conceived and designed the experiments, performed the experiments, analyzed the data, prepared figures and/or tables, authored or reviewed drafts of the article, and approved the final draft.
- Tuo Zhang conceived and designed the experiments, analyzed the data, prepared figures and/or tables, and approved the final draft.
- Fushen Luo conceived and designed the experiments, prepared figures and/or tables, and approved the final draft.
- Xiaofeng Yang conceived and designed the experiments, prepared figures and/or tables, and approved the final draft.
- Yadong Li analyzed the data, prepared figures and/or tables, and approved the final draft.
- Tonghui Yi performed the experiments, analyzed the data, authored or reviewed drafts of the article, and approved the final draft.

- Shuang Wu analyzed the data, authored or reviewed drafts of the article, and approved the final draft.
- Yanbing Wang analyzed the data, authored or reviewed drafts of the article, and approved the final draft.
- Yueping Zhu analyzed the data, authored or reviewed drafts of the article, and approved the final draft.
- Kun Zhao analyzed the data, authored or reviewed drafts of the article, and approved the final draft.

## Data Availability

Data is available at The Cancer Genome Atlas Program: TCGA-BRCA. Data is also available at the Gene Expression Omnibus (GEO) database: GSE20685.

## Supplemental Information

Supplemental information for this article can be found online at http://dx.doi.org/10.7717/peerj.19890#supplemental-information.

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
