# Peer review of "Exploration of prognostic genes associated with lymphangiogenesis in breast cancer based on transcriptomics and experimental verification"

_PeerJ, doi:10.7717/peerj.19890_

## Round 0.1 · original submission · Major Revisions

Please address all of the issues raised by the reviewers.

**Language Note:** The review process has identified that the English language must be improved. PeerJ can provide language editing services - please contact us at [email protected] for pricing (be sure to provide your manuscript number and title). Alternatively, you should make your own arrangements to improve the language quality and provide details in your response letter. – PeerJ Staff

Reviewer 1 ·

Basic reporting

The article contains numerous grammatical errors and typos.

Experimental design

I fail to see any clinical significance in this work.

Validity of the findings

The authors appear unaware that "BRCA" (the gene) and "breast cancer" (commonly abbreviated as "BC" in medical literature, not "BRCA" as in some databases) should not be conflated, as this creates confusion.

Additional comments

Breast cancer is not solely attributable to BRCA mutations.

Reviewer 2 ·

Basic reporting

The public datasets used in this study are appropriately cited. The bioinformatics tools used are appropriately cited.

The source code is well organized and readable.

Writing is mostly clear.

Figures are legible and clear.

The authors should more clearly link their results back to their initial lymphangiogenesis interest.

I was unable to extract the qPCR results, including Melt Curve Peak and Derivative Results xlsx, due to the file Paths being too long. Please shorten them and reupload.

Experimental design

The authors set out to investigate drivers of lymphangiogenesis in BRCA from the TCGA RNAseq dataset.
They compared normal versus tumor samples to identify differentially expressed genes, and overlapped those with lymphangiogenesis-related genes from annotated pathways in MSigDB. They split the tumor samples into high and low risk groups based on survival characteristics, and identified a subset of candidate lymphangiogenes-associated genes associated with low and high risk.

They further used lasso regression to subselect markers from this list.

They validated the directionality of their markers in tumor and normal cell lines.

Validity of the findings

The authors use rigorous, modern statistical and bioinformatic techniques to identify candidate genes.

They validate their markers in a small cohort of cell lines with qPCR. The authors make no inappropriate mechanistic claims and appropriately state their correlative findings.

In the results section title "Functional enrichment pathways of HRG and LRG and prognostic genes", please indicate the directionality of change for each pathway. The authors merely state that certain pathways, including TNFa-by NFKB, G2/M, and Interferon, are perturbed.

The results section "Analysis of prognostic genes with distinct immune cells" needs work. No consequences of infiltration are presented. Instead, an inference of different immune cell levels from bulk RNA sequencing is displayed. The authors should note that they are only making an inference. It is interesting to note the general downshift of all immune subtype markers in the high-risk group. The authors could state this more clearly.

The authors mention that CCL19 could be a prognostic marker, yet it is associated with lower risk and higher immune cell levels. How might it be used? Would it make patients candidates for immune checkpoint drugs, or are these drugs needed for the high-risk group with a generally more suppressed immune environment? Are the levels of CCR7 similarly elevated in the LRG?

Is there any evidence of increased key T regulatory markers in the HRG, such as high FOXP3 and CTLA4, and lower CD127? Admittedly, these can be imperfect markers.

The authors set out to investigate lymphangiogenesis and its role in BRCA promotion and metastasis, yet found that most immune cell populations, as inferred by RNAseq, are lower in the higher-risk group. Are there any more specific markers of lymphangiogenesis evident in either group?

Additional comments

Please rephrase the last sentence 366 for clarity: "The gene transcriptional tendencies were determined by the results of the bioinformatics analysis."

The authors should more clearly link their results back to their initial lymphangiogenesis interest in the discussion section.

·

Basic reporting

The manuscript describes the novel biomarkers in breast cancer using a bioinformatics approach. The authors need to extensively work on the rationale and motivation for each result section and tie the manuscript together. Please find detailed comments below:

General Comment: Improve figure resolution significantly.

Specific Comments:
1. Please provide references for “Breast cancer is one of the most common malignancies, and in some regions, it has surpassed lung cancer in incidence rates.”
2. Please provide references for “lung cancer in incidence rates. In China, the incidence of breast cancer has been increasing annually, likely due to changes in lifestyle, reproductive patterns, and improved diagnostic capabilities.”
3. Is there any correlation between BRCA1/2 mutations and lymphangiogenesis? If yes/no, are the authors trying to bridge this gap in research? Please include details in the introduction.
4. Please explain the abbreviations DOWN, UP, NOT, and BC in the Figure 1 legend.
5. Figure 1C shows LRG, and the legend shows L-RG. Both abbreviations have different meanings. Please use correct abbreviations.
6. Please explain the different colors used in Figure 1F. The gene names are not readable in this figure.
7. Please provide a concluding statement and a transitioning statement for every result section.
8. What is a PH assumption test? Why is it needed? Please give a rationale.
9. The authors state, “> 0.05) (Table S4). Through univariate Cox regression analysis, 4 prognosis-related genes (ZIC2, CD24, CEBPD, and CCL19) were finally obtained. Were these obtained from 98 genes? If yes, please clarify.
10. Please explain each figure in Figure 2D panel. The authors drew the conclusion from Figure 2D without explaining the results completely.
11. Please explain explicitly the training and the validation dataset, and then compare the results.
12. The authors state, “With the application of univariate Cox analysis (p < 0.05) and PH assumption test (p > 0.05), among the risk score and the risk score, age, N, and M met the requirements (Fig. 3A, Table S5).” Please explain how Fig. 3A meets the requirements.
13. The authors state, “A nomogram was generated (Fig. 3C). Meanwhile, the slopes of the nomogram-predicted survival probabilities in calibration curves were close to 1 (Fig. 3D).” What does this imply?
14. For the result section “Functional enrichment pathways of HRG and LRG and prognostic genes,” what is the reference used? Are these pathways enriched in HRG or LRG? What does the X axis in Figure 4C stand for?
15. What were the criteria for selecting the breast cancer cell lines?
16. The authors state, “The gene transcriptional tendencies were determined by the results of the bioinformatics analysis.” The result consists of both RT-qPCR and bioinformatics data. Please explain.
17. The authors talk about BRCA1/2 mutations in the introduction, and neither the results nor the discussion tie in with this. The authors need to clarify their thought process.
18. Please explain role of CCL19, CEBPD, ZIC2, CD24 in lymphangiogenesis in the discussion section.

Experimental design

Please refer to basic reporting section

Validity of the findings

Please refer to basic reporting section

---

## Round 0.2 · accepted · Accept

Thanks for addressing all reviewers comments and congratulations.

Reviewer 2 ·

Basic reporting

I am now able to access all supplemental files and raw data. Everything is well organized and clearly labeled.

The manuscript reads more clearly.

Experimental design

The results section discussing Figure 4 is much improved.

Validity of the findings

The discussion section reads much more coherently and is much clearer in discussing the implications of their findings, suggesting the low-risk group may be more amenable to immunotherapy.

The discussion section on CCL19 is much improved.

Additional comments

This is an interesting window into the differing immune landscapes between low and high-risk BC groups.

·

Basic reporting

-

Experimental design

-

Validity of the findings

-

Additional comments

The authors have addressed all the concerns.